# A Radiation-Regulated Dynamic Maximum Light Use Efficiency for Improving Gross Primary Productivity Estimation

Zhiying Xie [1], Cenliang Zhao [2,*], Wenquan Zhu [2], Hui Zhang [2] and Yongshuo H. Fu [1]

1   College of Water Sciences, Beijing Normal University, Beijing 100875, China
2   State Key Laboratory of Remote Sensing Science, Faculty of Geographical Science, Beijing Normal University, Beijing 100875, China
*   Correspondence: zhao.cl@mail.bnu.edu.cn

**Abstract:** The light use efficiency (LUE) model has been widely used in regional and global terrestrial gross primary productivity (GPP) estimation due to its simple structure, few input parameters, and particular theoretical basis. As a key input parameter of the LUE model, the maximum LUE ($\mathcal{E}_{\max}$) is crucial for the accurate estimation of GPP and to the interpretability of the LUE model. Currently, most studies have assumed $\mathcal{E}_{\max}$ as a universal constant or constants depending on vegetation type, which means that the spatiotemporal dynamics of $\mathcal{E}_{\max}$ were ignored, leading to obvious uncertainties in LUE-based GPP estimation. Using quality-screened daily data from the FLUXNET 2015 dataset, this paper proposed a photosynthetically active radiation (PAR)-regulated dynamic $\mathcal{E}_{\max}$ (*PAR-$\mathcal{E}_{\max}$*, corresponding model named PAR-LUE) by considering the nonlinear response of vegetation photosynthesis to solar radiation. The PAR-LUE was compared with static $\mathcal{E}_{\max}$-based (MODIS and EC-LUE) and spatial dynamics $\mathcal{E}_{\max}$-based (D-VPM) models at 171 flux sites. Validation results showed that (1) $R^2$ and RMSE between PAR-LUE GPP and observed GPP were 0.65 (0.44) and 2.55 (1.82) g C m$^{-2}$ MJ$^{-1}$ d$^{-1}$ at the 8-day (annual) scale, respectively; (2) GPP estimation accuracy of PAR-LUE was higher than that of other LUE-based models (MODIS, EC-LUE, and D-VPM), specifically, $R^2$ increased by 29.41%, 2.33%, and 12.82%, and RMSE decreased by 0.36, 0.14, and 0.34 g C m$^{-2}$ MJ$^{-1}$ d$^{-1}$ at the annual scale; and (3) specifically, compared to the static $\mathcal{E}_{\max}$-based model (MODIS and EC-LUE), PAR-LUE effectively relieved the underestimation of high GPP. Overall, the newly developed *PAR-$\mathcal{E}_{\max}$* provided an estimation method utilizing a spatiotemporal dynamic $\mathcal{E}_{\max}$, which effectively reduced the uncertainty of GPP estimation and provided a new option for the optimization of $\mathcal{E}_{\max}$ in the LUE model.

**Keywords:** gross primary production (GPP); light use efficiency (LUE); photosynthetically active radiation (PAR); dynamic maximum LUE

## 1. Introduction

As the largest component of the terrestrial carbon cycle [1–3], the accurate estimation of terrestrial gross primary productivity (GPP) is vital for understanding global carbon cycle processes, climate changes, and ecosystem services [1,4–6]. To date, GPP estimation is difficult, as no direct measures are available at regional and global scales. The eddy covariance flux tower provides a direct measure of carbon dioxide ($CO_2$) between the land surface and the atmosphere that can be used for indirect GPP estimation [7]. Flux-based GPP has been widely used as reference data for calibrating and validating GPP models [8–12]. With the development of GPP estimation theories and methods in recent decades, researchers have developed many remote sensing models for GPP estimation [3,13,14].

Light use efficiency (LUE) models have been widely used to estimate terrestrial GPP at regional and global scales due to their theoretical basis, few parameters, and high practicality [15]. Since Monteith [16] proposed the concept of LUE, following the theoretical basis of LUE [17], researchers have optimized the input parameters for calculating LUE

and developed dozens of LUE models [2]. As a key parameter of the LUE model, the maximum LUE ($LUE_{max}$, same as $\mathcal{E}_{max}$) is crucial for the accurate estimation of GPP and for the interpretability of the LUE model. The $\mathcal{E}_{max}$ in existing LUE models can be roughly divided into three main categories: (1) global constant $\mathcal{E}_{max}$, such as the constant $\mathcal{E}_{max}$ used in C-Fix [18] and EC-LUE [12]; (2) constant $\mathcal{E}_{max}$ for each type, such as the constant $\mathcal{E}_{max}$ varying from vegetation types in MOD17 [17] and VPRM [19], the constant $\mathcal{E}_{max}$ for C3 and C4 in VPM [10,20,21] and TEC [22], the constant $\mathcal{E}_{max}$ for sunlit and shaded leaves in TL-LUE [23] and DTEC [24], and the constant $\mathcal{E}_{max}$ for different phenological stages in TS-LUE [25]; and (3) dynamic $\mathcal{E}_{max}$, such as the cloudiness index-regulated dynamic $\mathcal{E}_{max}$ in CFlux [26], CI-LUE [27], and CI-EF [28], and the spatial dynamic $\mathcal{E}_{max}$ based on the enhanced vegetation index (EVI) and visible albedo [29].

With the development of LUE models, an increasing number of researchers have considered $\mathcal{E}_{max}$ as a dynamic value rather than a constant. Dynamic $\mathcal{E}_{max}$ is more consistent with vegetation physiology, and studies have proven that in their own study area, dynamic $\mathcal{E}_{max}$ performed better than static $\mathcal{E}_{max}$ in GPP estimation [30,31]. Although researchers have proposed methods to estimate dynamic $\mathcal{E}_{max}$, the nonlinear response of vegetation photosynthesis to solar radiation variation was rarely considered in these studies. As the energy source of vegetation photosynthesis, solar radiation variation directly regulated the vegetation $\mathcal{E}_{max}$. Most LUE models implied a linear relationship between GPP and PAR, which apparently ignored the saturation of vegetation photosynthesis to solar radiation under high solar radiation. In fact, vegetation photosynthesis varies with the dynamics of solar radiation [32]. For the low radiation situation, especially when vegetation photosynthesis is limited only by radiation, photosynthesis would increase linearly with increasing radiation. For the high radiation situation (plentiful radiation), photosynthesis would become radiation-saturated and no longer respond to the changes in radiation supply [32–36].

Considering the nonlinear response of vegetation photosynthesis to solar radiation [37], this paper proposed a photosynthetically active radiation (PAR)-regulated dynamic $\mathcal{E}_{max}$ (PAR-$\mathcal{E}_{max}$, corresponding model named PAR-LUE) by using the quality-screened daily GPP and PAR data from the FLUXNET 2015 dataset. The performances of PAR-LUE in GPP estimation were evaluated based on the observed GPP and other LUE-based GPP estimation results.

## 2. Data and Preprocessing

### 2.1. FLUXNET Data

The FLUXNET 2015 dataset (https://fluxnet.fluxdata.org/data/FLUXNET2015-dataset/ (accessed on 1 August 2022)) includes multiple temporal scales (e.g., half-hourly, hourly, daily, weekly, monthly, and yearly) of observations, which contains the flux data of carbon, water, and energy collected from 212 sites around the globe. Data were quality-controlled and processed using uniform methods to improve the consistency and intercomparability across sites [38,39]. In this paper, a total of 171 sites (1104 site-years) with high quality data ("NEE_QC" > 0.75) were selected from the FLUXNET 2015 dataset (Figure 1), and the variables of the daily "GPP_NT_VUT_MEAN" and " SW_IN_F" were selected as the reference daily GPP (g C $m^{-2}$ $d^{-1}$) and shortwave radiation (SR; W $m^{-2}$). In addition, the daily PAR was calculated using the site observed shortwave radiation according to the empirical formula (i.e., PAR = 0.45 × SR × 0.0864; MJ $m^{-2}$). More details about the FLUXNET 2015 dataset can be found in Pastorello, Trotta [38].

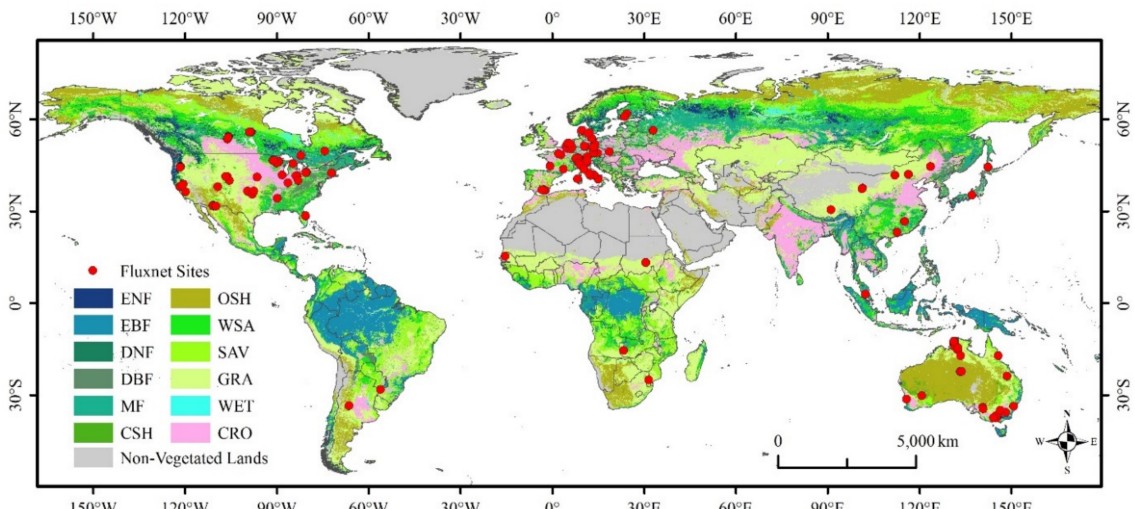

**Figure 1.** Spatial distribution of selected FLUXNET 2015 sites. (Background is the map of the MCD12Q1 in 2014; CRO: cropland (19 sites), CSH: closed shrub (2 sites), DBF: deciduous broadleaf forest (22 sites), DNF: deciduous needleleaf forests (0 site), EBF: evergreen broad-leaf forest (11 sites), ENF: evergreen needleleaf forest (44 sites), MF: mixed forest (8 sites), GRA: grassland (30 sites), OSH: open shrub (11 sites), SAV: savanna (7 sites), WSA: woody savanna (6 sites), WET: wetland (11 sites)).

### 2.2. MODIS Data

Daily MODIS surface reflectance products (MCD43A4 Version 6) and 8-day composite MODIS GPP products (MOD17A2H Version 6) with 500 m resolution were used in this study. The MCD43A4 and MOD17A2H for each carbon flux site were downloaded from the NASA MODIS/VIIRS Land Product Subsets (https://modis.ornl.gov/globalsubset/ (accessed on 1 August 2022)). All MODIS data were filtered according to their quality label. MOD17A2H was used as one of the comparison data to evaluate the performance of PAR-LUE in GPP estimation, and MCD43A4 was used to calculate visible albedo, EVI, and land surface water index (LSWI). Albedo, EVI, and LSWI were calculated as follows:

$$\text{Albedo}_{\text{visible}} = 0.331 R_{Red} + 0.424 R_{Blue} + 0.246 R_{Green} \tag{1}$$

$$EVI = 2.5 \times \frac{R_{NIR} - R_{Red}}{R_{NIR} + 6R_{Red} - 7.5R_{Blue} + 1} \tag{2}$$

$$LSWI = \frac{R_{NIR} - R_{SWIR}}{R_{NIR} + R_{SWIR}} \tag{3}$$

where $R_{Blue}$, $R_{Red}$, $R_{NIR}$, and $R_{SWIR}$ are the reflectances of the blue, red, near infrared (NIR), and shortwave infrared (SWIR) bands, respectively.

## 3. Methods

### 3.1. PAR-LUE

The common structure of the LUE model can be formulated as follows:

$$GPP = PAR \times FPAR \times \mathcal{E}_{max} \times f(Ts) \times f(Ws) \tag{4}$$

where PAR is photosynthetically active radiation, FPAR is the fraction of absorbed PAR, $\mathcal{E}_{max}$ is maximal light use efficiency, and $f(Ts)$ and $f(Ws)$ are the scaled environmental stress indices of temperature and water on LUE, respectively.

Referring to the existing research results [10,21], FPAR, f (Ts), and f (Ws) were calculated as follows:

$$FPAR = (EVI - 0.1) \times 1.25 \tag{5}$$

$$T_S = \frac{(T - T_{min})(T - T_{max})}{(T - T_{min})(T - T_{max}) - (T - T_{opt})^2} \tag{6}$$

$$W_S = \frac{1 + LSWI}{1 + LSWI_{max}} \tag{7}$$

where $T$ is the daily temperature; $T_{min}$, $T_{max}$, and $T_{opt}$ are 0 °C, 40 °C, and 20 °C, respectively; and $LSWI_{max}$ is the maximal LSWI in the growing season. Here, the growing season is defined according to the date of 75 days before and after the date of maximal EVI (i.e., [date_EVI$_{max}$–75, date_EVI$_{max}$ + 75]).

In the PAR-LUE model, we proposed a PAR-based method to calculate dynamic $\mathcal{E}_{max}$ (i.e., $PAR$-$\mathcal{E}_{max}$, and $PAR$-$\mathcal{E}_{max} = f$ (PAR)). Considering the nonlinear response of vegetation photosynthesis to PAR, we developed the PAR-LUE model based on two hypothesises. First, under the ideal condition that vegetation was only related to the PAR and unconstrained by other biotic and abiotic conditions (i.e., FPAR, f ($Ts$), and f ($Ws$) are equal to 1), GPP$_i$ can be represented as the product of PAR and $\mathcal{E}_{max}$ (8). Under the optimal temperature and water conditions (i.e., T = T$_{opt}$, LSWI = LSWI$_{max}$), the f ($Ts$) and f ($Ws$) are equal to 1. For the FPAR, the maximum value of measured [40] and remotely sensed FPAR (e.g., for the formula (5), FPAR = 1 when EVI ≥ 0.9) are close or equal to 1. Second, the maximum GPP (GPP$_{max}$) under different levels of PAR meets the ideal conditions (i.e., GPP$_i$ = GPP$_{max}$).

Using the quality-screened daily GPP and PAR data from the FLUXNET 2015 dataset, the maximum value of GPP corresponding to PAR (PAR was sampled with a step of 1 MJ m$^{-2}$) was sampled within the subrange of 1 ± 0.25 MJ m$^{-2}$. Then, the sampled PAR and GPP$_{max}$ were fitted using cubic polynomial (9). Finally, combining Formulas (8) and (9), $PAR$-$\mathcal{E}_{max}$ was calculated with Formula (10), and the corresponding model was named PAR-LUE.

$$GPP_i = PAR \times \mathcal{E}_{max} = PAR \times f\ (PAR) \tag{8}$$

$$GPP_{max} = a \times PAR^3 + b \times PAR^2 + c \times PAR \tag{9}$$

$$PAR\text{-}\mathcal{E}_{max} = a \times PAR^2 + b \times PAR + c \tag{10}$$

where GPP$_i$ is the GPP under ideal conditions, and $a$, $b$, and $c$ are fitting parameters of the cubic polynomial.

As shown in Figure 2, the orange fit curve indicated the vegetation GPP$_{max}$ under different PAR levels, and the points below the fit curve indicated the true GPP constrained by vegetation physiology and environmental factors. It is important to note that PAR-$\mathcal{E}_{max}$ is the $\mathcal{E}_{max}$ of all vegetation in the ideal condition. For the differences in $\mathcal{E}_{max}$ among vegetation types, it is expected that the EVI-based FPAR can regulate those differences under the framework of the LUE model.

### 3.2. Reference LUE Model

In this paper, static $\mathcal{E}_{max}$- and spatial dynamic $\mathcal{E}_{max}$-based LUE models were built as reference models (named EC-LUE and D-VPM, respectively) to examine the performances of the newly developed PAR-$\mathcal{E}_{max}$-based model (PAR-LUE) under the same LUE model framework. Among those models, only $\mathcal{E}_{max}$ is different. Specifically, the static $\mathcal{E}_{max}$ is 2.14 g C m$^{-2}$ MJ$^{-1}$ [12], while the spatial dynamic $\mathcal{E}_{max}$ (named RS-$\mathcal{E}_{max}$) is calculated according to the EVI and albedo [29]:

$$\begin{cases} \mathcal{E}_{max} = \exp(1.428 MaxE - 6.295 MinVa - 1.211) & , MaxE > 0.07 \\ \mathcal{E}_{max} = 0, & MaxE \leq 0.07 \end{cases} \tag{11}$$

where *MaxE* and *MinVa* are the maximal EVI and minimal visible albedo in the growing season, respectively.

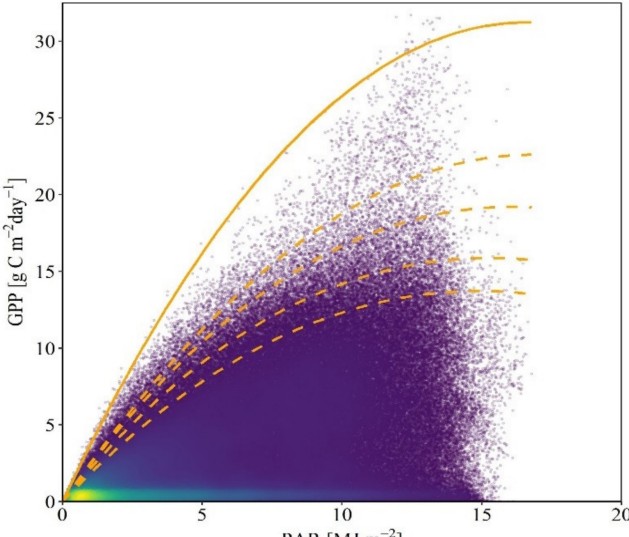

**Figure 2.** Schematic of *PAR-$\mathcal{E}_{max}$* based on the relationship between daily observed GPP and PAR. (The color from purple to yellow indicates that the point density increases gradually. The orange fit curve indicates the vegetation GPPmax at different PAR levels, and the dashed curves from top to bottom are fitted curves at the 99th, 98th, 95th, and 90th percentiles of GPP$_{max}$. The fitted curve indicates the unconstrained ideal GPP, while the points below the curve indicate the true GPP constrained by vegetation physiology and environmental factors.)

### 3.3. Accuracy Evaluation

Two criteria were used here to evaluate model performance, including the determination coefficient ($R^2$) and root mean square error (RMSE). In addition to the validation with observed GPP from FLUXNET sites, the performances of the PAR-LUE model in GPP estimation were compared with that of the MOD17 algorithm, EC-LUE and D-VPM from multiple dimensions, such as overall accuracy (8-day and annual), accuracy of each vegetation type and the seasonal dynamics in typical sites.

## 4. Results

### 4.1. Comparison of Different $\mathcal{E}_{max}$

Different PAR-$\mathcal{E}_{max}$ calculated usingthe different GPP$_{max}$ sample percentiles have similar performances in GPP estimation (Table 1). With the decrease in the GPP$_{max}$ sample percentile, the $R^2$ between the PAR-LUE-estimated GPP and observed GPP slightly increased, while the RMSE obviously increased (hereafter, *PAR-$\mathcal{E}_{max}$* was calculated according to GPP$_{max}$). At all selected flux sites, the spatiotemporal dynamic ranges of daily *PAR-$\mathcal{E}_{max}$* and RS-$\mathcal{E}_{max}$ were 1.86–3.85 g C m$^{-2}$ MJ$^{-1}$ and 0.73–4.39 g C m$^{-2}$ MJ$^{-1}$, respectively (Figure 3). Obviously, the variation range of RS-$\mathcal{E}_{max}$ was larger than that of *PAR-$\mathcal{E}_{max}$*, and both dynamic $\mathcal{E}_{max}$ contained a constant value of 2.14 g C m$^{-2}$ MJ$^{-1}$.

*PAR-$\mathcal{E}_{max}$* showed significant seasonal dynamics, which presented as a "U" shaped variation in a natural year (Figure 4). The seasonal trends of *PAR-$\mathcal{E}_{max}$* were relatively similar at the 10 typical vegetation sites, and their annual minimal *PAR-$\mathcal{E}_{max}$* values were close to 2.14 g C m$^{-2}$ MJ$^{-1}$. However, although RS-$\mathcal{E}_{max}$ showed spatial and interannual variations, there was no seasonal dynamic.

**Table 1.** *PAR-$\mathcal{E}_{max}$* estimation coefficients and GPP estimation accuracy under different sampling percentiles.

| Percentile | a | b | c | $R^2$ | RMSE |
|---|---|---|---|---|---|
| 100th | 0.00030 | −0.12376 | 3.84951 | 0.61202 | 2.8979 |
| 99th | −0.00039 | −0.06768 | 2.59194 | 0.61019 | 2.9812 |
| 98th | 0.00052 | −0.09087 | 2.52062 | 0.61343 | 3.1780 |
| 95th | 0.00073 | −0.09035 | 2.24867 | 0.61492 | 3.4993 |
| 90th | 0.00033 | −0.07156 | 1.91375 | 0.61463 | 3.7946 |

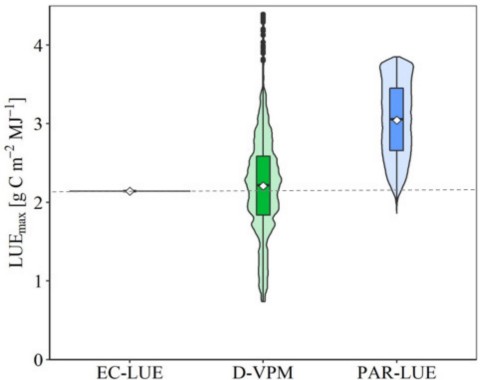

**Figure 3.** Variation range comparison of three daily $\mathcal{E}$max at all selected flux sites. (The top and bottom edges of the box are the 75% and 25% quartiles, respectively; the short horizontal line and the small square in the middle of the box are the median and mean, respectively; the shape of the violin displays frequencies of values.)

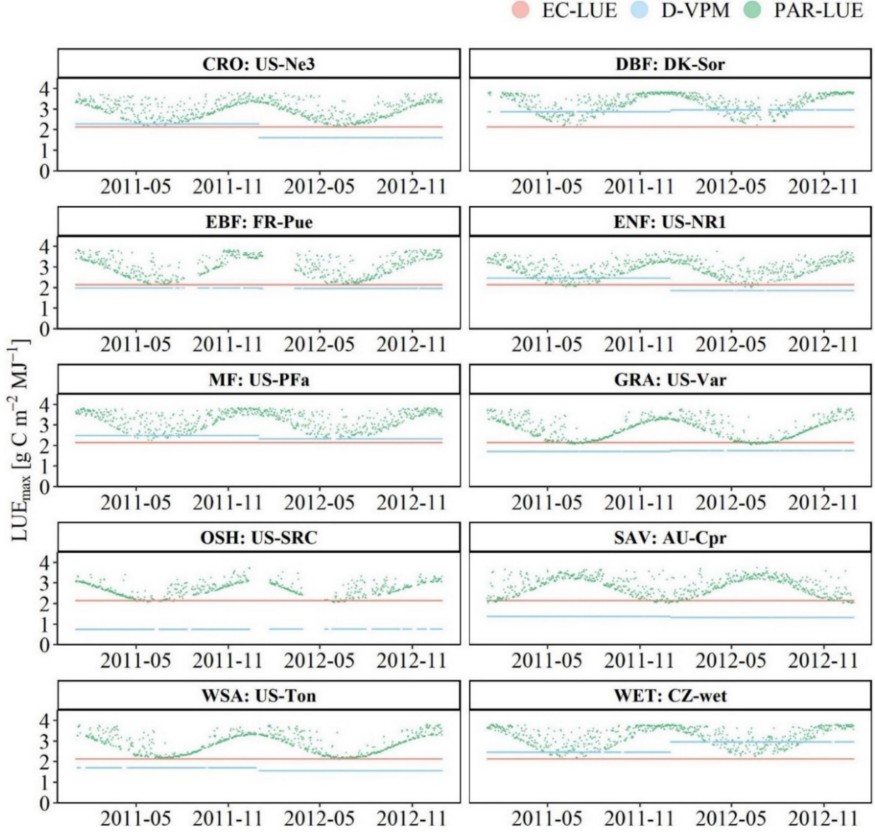

**Figure 4.** Seasonal dynamics comparison of three $\mathcal{E}_{max}$ at typical sites. Ten sites from different vegetation types were selected and shown in two adjacent years (2011–2012) to exhibit the differences in seasonal variation among the three $\mathcal{E}_{max}$.

### 4.2. Comparison of GPP Estimation

The overall estimation accuracy of PAR-LUE GPP was better than that of MODIS GPP, EC-LUE GPP, and D-VPM GPP (Figure 5). Compared with MODIS GPP, EC-LUE GPP, and D-VPM GPP at the 8-day scale, the $R^2$ between PAR-LUE GPP and observed GPP increased by 12.07%, 1.56%, and 8.33%, and the RMSE decreased by 0.18, 0.09, and 0.41 g C m$^{-2}$ MJ$^{-1}$ d$^{-1}$, respectively. At the annual scale, $R^2$ increased by 29.41%, 2.33%, and 12.82%, and the RMSE decreased by 0.36, 0.14, and 0.34 g C m$^{-2}$ MJ$^{-1}$ d$^{-1}$, respectively. Although the GPP estimation accuracies of PAR-LUE and EC-LUE were closer in $R^2$ and RMSE, EC-LUE obviously underestimated the high GPP. Overall, PAR-LUE showed better performances than the reference LUE models in GPP estimation, especially in reducing the underestimation of high GPP. From the performances of LUE models in different vegetation types (Figure 6), the GPP estimation accuracy of PAR-LUE was generally comparable to that of EC-LUE and D-VPM in most types.

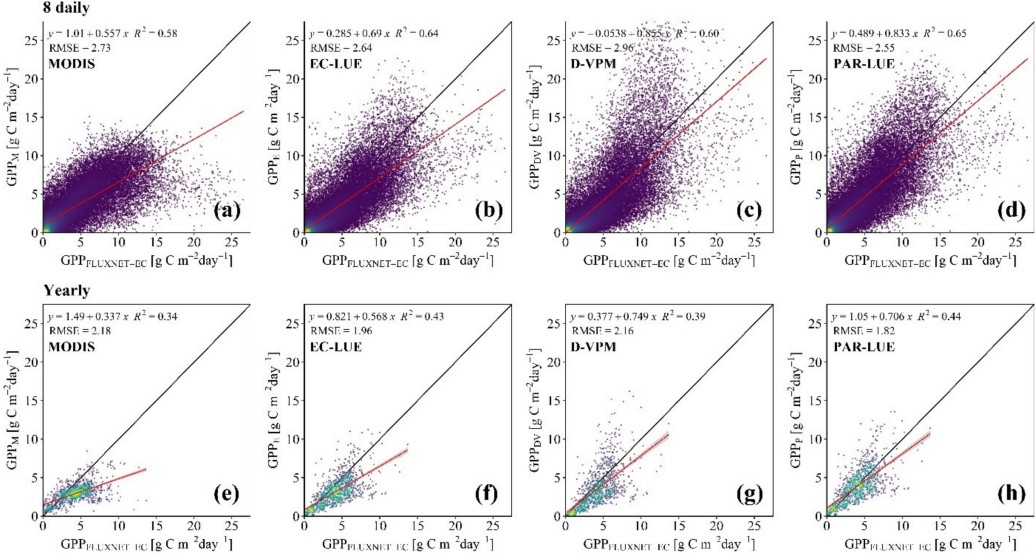

**Figure 5.** Accuracy comparison of GPP estimated from different LUE models. (**a**–**d**) show the 8 daily results, and (**e**–**h**) show the yearly results.

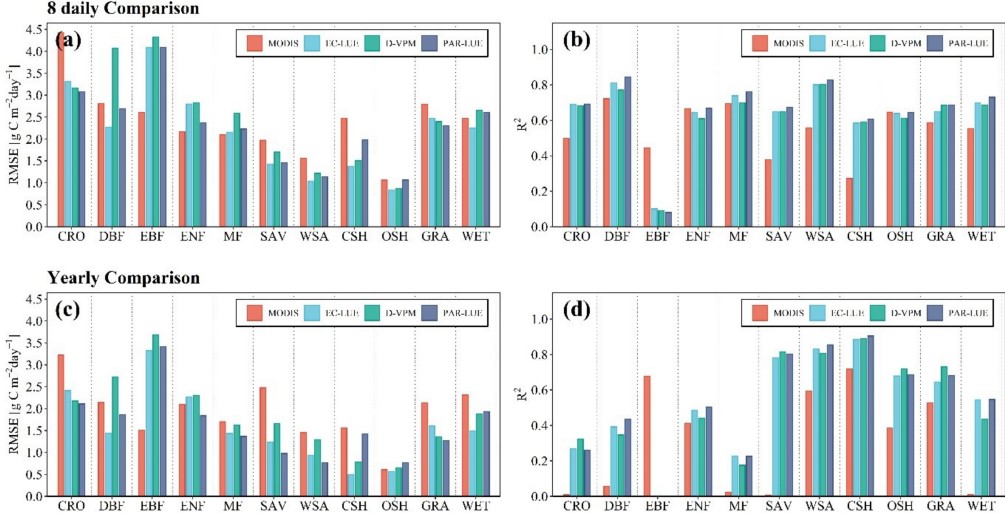

**Figure 6.** Accuracy comparison of GPP estimated from different LUE models in different vegetation types. (**a**) Comparison of RMSE at the 8-day scale, (**b**) comparison of $R^2$ at the 8-day scale, (**c**) comparison of RMSE at the yearly scale, and (**d**) comparison of $R^2$ at the yearly scale.

The comparison at typical sites in the Northern Hemisphere showed that PAR-LUE, EC-LUE, and D-VPM were all in good agreement with the observed GPP in characterizing seasonal dynamics (Figure 7). In the majority of vegetation types, the PAR-LUE GPP was in better agreement with the observed GPP, which is evidenced by a smaller RMSE and a closer regression slope to 1. For example, at the evergreen coniferous forest site (Figure 7c), although the GPP estimated using the three LUE models showed very high agreement with the observed GPP ($R^2 \geq 0.93$), the PAR-LUE-estimated GPP was closer to the observed GPP throughout the growing season (smaller RMSE and closer slope to 1); at the closed shrub site (Figure 7g), the PAR-LUE GPP was closer to the observed GPP during the rising and falling phases of the growing season.

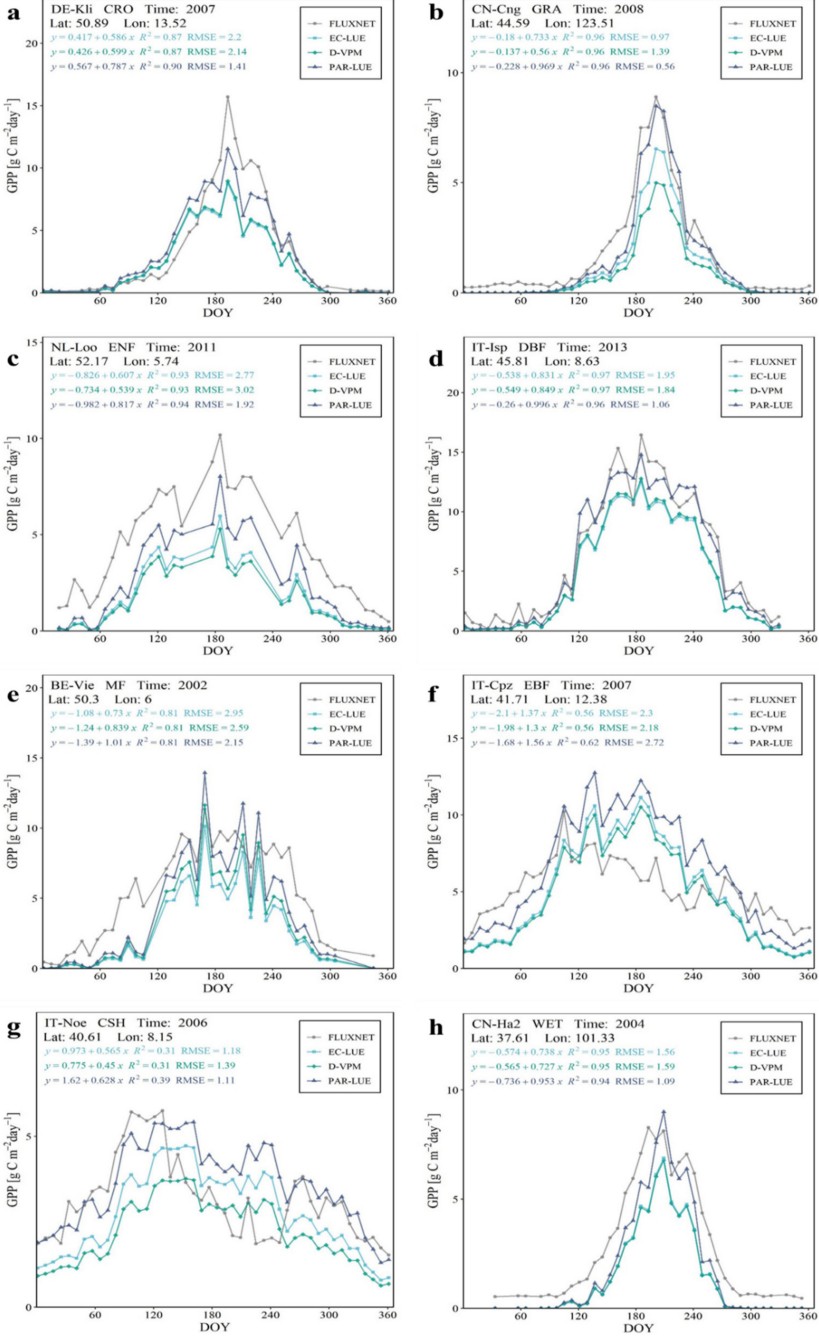

**Figure 7.** Comparison of GPP seasonal variation at typical sites. Eight site-year samples ((**a–h**), Lat: latitude; Lon: longitude; see Figure 1 for vegetation types) with similar or concurrently higher $R^2$ values were selected to exhibit the better performance of the PAR-LUE model in further reducing the RMSE.

## 5. Discussion

As the direct energy source for vegetation photosynthesis, solar radiation directly determines the light use efficiency of vegetation. Therefore, $\mathcal{E}_{max}$ should have corresponding spatiotemporal dynamics to the background of spatiotemporal solar radiation variation. The spatiotemporal dynamics of $\mathcal{E}_{max}$ were the result of the long-term adaptation of vegetation to variations in solar radiation (especially seasonal variations), and vegetation usually has different $\mathcal{E}_{max}$ values under different radiation conditions. The seasonal dynamics of *PAR*-$\mathcal{E}_{max}$ presented in this paper show a "U"-shaped trend, with a large value in spring and autumn and a small value in summer. During vegetation green-up, vegetation has a large $\mathcal{E}_{max}$ to make full use of the limited PAR and thus promote vegetation growth and development. However, during the peak growing season, solar radiation is more abundant, and vegetation photosynthesis tends to be saturated (high solar radiation will even reduce photosynthesis) and has a relatively small $\mathcal{E}_{max}$. There were similar explanations in the studies of Chapin and Matson [32] and Mõttus and Sulev [34]. The study of Propastin and Ibrom [41] in a tropical rainforest found that the LUE model would overestimate vegetation GPP under high radiation conditions if the saturation effect of vegetation photosynthesis on solar radiation is not considered. In addition, some studies showed that $\mathcal{E}_{max}$ was different in clear and cloudy skies [37], as well as in sunlit and shaded leaves [31], which partly explains the influence of PAR on $\mathcal{E}_{max}$. Some cloudiness indices that regulate $\mathcal{E}_{max}$ used in CFlux [26], CI-LUE [27], and CI-EF [28] can also be considered a sort of radiation-regulated $\mathcal{E}_{max}$ because the cloudiness index was calculated based on PAR [42].

In this paper, a cubic polynomial function was used to calculate *PAR*-$\mathcal{E}_{max}$, which is simple to calculate and easy to fit. The fitted curve can effectively characterize the relationship between PAR and GPP in the actual range of PAR variation, and the shape of the fitted curve is consistent with existing studies [32–34,36]. The $\mathcal{E}_{max}$ can be estimated based on different data and methods, and there are differences in their physical meanings [43]. Some studies estimated $\mathcal{E}_{max}$ based on the flux data observed during the peak growing season [44–46], which obtained the specific $\mathcal{E}_{max}$ under radiation saturation. In terms of the full growing season, $\mathcal{E}_{max}$ of the peak growing season is only a special case of its seasonal dynamic changes. For example, the specific constant $\mathcal{E}_{max}$ used in the EC-LUE model was comparable to the seasonal minimum of *PAR*-$\mathcal{E}_{max}$ (Figure 3). In the comparison of $\mathcal{E}_{max}$ defined by different studies, extra attention needs to be paid to their essential meanings. For example, the study of Zhang and Xiao [43] indicated that the daily $\mathcal{E}_{max}$ exhibits less variation across biome types and seasons, which contradicts the spatiotemporal dynamics of *PAR*-$\mathcal{E}_{max}$. It is important to note that the $\mathcal{E}_{max}$ defined by Zhang and Xiao [43] contains the FPAR, while the FPAR in *PAR*-$\mathcal{E}_{max}$ is assumed to be 1. In terms of seasonal trends, PAR-$\mathcal{E}_{max}$ was consistent with the reference LUE in the study of Zhang and Xiao [43].

Dynamic $\mathcal{E}_{max}$-based PAR-LUE performed better in GPP estimation than that of constant $\mathcal{E}_{max}$-based MODIS GPP and EC-LUE. In terms of GPP estimation accuracy alone ($R^2$ and RMSE), EC-LUE was comparable to PAR-LUE in GPP estimation. However, the PAR-LUE mitigated the underestimation of high GPP, which is a nonnegligible contribution to an accurate estimate of total annual GPP. The constant $\mathcal{E}_{max}$ in the EC-LUE model, as a special case of dynamic $\mathcal{E}_{max}$, is one of the reasons for its comparable ability to estimate GPP (especially $R^2$) with the PAR-LUE model. On the one hand, the GPP values at the beginning and end of the growing season were relatively small, so only a minor difference existed between the GPP estimated by the constant and dynamic $\mathcal{E}_{max}$. For example, in the comparison of the seasonal variation in GPP at typical sites in the Northern Hemisphere (Figure 7), EC-LUE and PAR-LUE showed similar $R^2$ values, but the seasonal dynamics of PAR-LUE GPP were closer to the observed GPP. On the other hand, the accuracy of other input parameters in the LUE model may inhibit the ability of dynamic $\mathcal{E}_{max}$ to improve the accuracy of GPP estimation. In addition, compared with the spatial dynamics of $\mathcal{E}_{max}$-based D-VPM, PAR-LUE has two advantages in addition to the improvement in precision. First, PAR drives vegetation photosynthesis more directly than albedo, the $\mathcal{E}_{max}$ constructed by PAR is more theoretical than that of albedo, and *PAR*-$\mathcal{E}_{max}$ has both

spatial and temporal dynamic characteristics. Second, the estimation of spatial dynamic *RS-*$\mathcal{E}_{max}$ requires remote sensing data in a whole growing season (it requires the maximum EVI and minimum albedo of the whole growing season), which limits its application in near-real-time GPP estimation, while *PAR-*$\mathcal{E}_{max}$ is not affected by this.

The accuracy validation results showed the reasonableness and reliability of *PAR-*$\mathcal{E}_{max}$. However, PAR-LUE still has room for improvement. First, the accurate estimation of *PAR-*$\mathcal{E}_{max}$ requires a large number of high-quality observations. Limited by site representativeness and data quality, *PAR-*$\mathcal{E}_{max}$ may have some errors in the specific values. A large number of high-quality observations will help to improve the performance of the PAR-LUE model. Second, *PAR-*$\mathcal{E}_{max}$ characterizes the $\mathcal{E}_{max}$ of all vegetation, and it was thought that the EVI-based FPAR could constrain the differences in *PAR-*$\mathcal{E}_{max}$ across vegetation types in the PAR-LUE model. However, from the performances of PAR-LUE in estimating vegetation GPP (Figures 5–7), FPAR was able to partly characterize differences in vegetation types, but its ability to constrain *PAR-*$\mathcal{E}_{max}$ was still limited. Obviously, the *PAR-*$\mathcal{E}_{max}$ of different vegetation types can be obtained based on vegetation type data, but it will be limited by the quality of vegetation type data, and it is difficult to characterize the spatially continuous variation in terrestrial vegetation with vegetation type data. It is hoped that further research can introduce a factor that can characterize the spatially continuous variation in vegetation photosynthetic capacity in PAR-LUE to improve the theory of the PAR-LUE model and the accuracy of GPP estimation. Finally, the effect of other input parameters of the LUE model on the accuracy of GPP estimation needs to be further analyzed to reduce the interference from other input parameters on the contribution of $\mathcal{E}_{max}$ parameter optimization to the improvement of GPP estimation.

## 6. Conclusions

Considering the nonlinear response of vegetation photosynthesis to solar radiation, we proposed a new $\mathcal{E}_{max}$ with spatiotemporal dynamics (*PAR-*$\mathcal{E}_{max}$, model denoted PAR-LUE), with using the daily PAR and GPP data observed from the flux tower, based on the assumption that GPP is only determined by PAR and $\mathcal{E}_{max}$ under ideal conditions. Flux data-based validation results showed that the accuracy of PAR-LUE-estimated GPP was better than that of constant and spatially varied $\mathcal{E}_{max}$-based models. The PAR-LUE was suitable for remote sensing based GPP estimation of most vegetation types at regional and global scales. Overall, the newly developed *PAR-*$\mathcal{E}_{max}$ provided an estimation method of spatiotemporal dynamic $\mathcal{E}_{max}$, which effectively reduced the uncertainty of GPP estimation and provided a new option for the optimization and development of dynamic $\mathcal{E}_{max}$ in the LUE model.

**Author Contributions:** Conceptualization, Z.X., C.Z., W.Z. and Y.H.F.; methodology, Z.X. and C.Z.; writing—original draft preparation, Z.X. and C.Z.; writing—review and editing, Z.X., C.Z., H.Z., W.Z. and Y.H.F.; visualization, C.Z. and H.Z. All authors have read and agreed to the published version of the manuscript.

**Funding:** This study is funded by the National Key Research and Development Program of China (Grant No. 2020YFA0608504) and the Second Tibetan Plateau Scientific Expedition and Research Program (STEP) (Grant No. 2019QZKK0606).

**Data Availability Statement:** Data is available upon request from the corresponding author.

**Conflicts of Interest:** The authors declare no conflict of interest.

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
