# Peer review of "A Radiation-Regulated Dynamic Maximum Light Use Efficiency for Improving Gross Primary Productivity Estimation"

_remotesensing, doi:10.3390/rs15051176_

Round 1
Reviewer 1 Report
The presentation of the study is clear and technically comprehensive, providing sufficient information for understanding of the methodological approach, data sources and interpretation of results. The study clearly compare 2 modelling approaches and provides potentially better solution in comparison to the reference approach selected in the article. However, quality of the field measurement data (gas flux data) are crucial for studies aimed at verification of the model, and quality of these data are not reflected sufficiently in the manuscript, including seasonality, management, moisture regime. The types of vegetation in the selected sites and selection of certain types of vegetation (if any selection is done) are not sufficiently explained, too. I'm not expert in climate data, but according to our experience selection of data source for PAR is crucial, probably, the quality of the applied (or calculated) PAR data can be evaluated further in the manuscript. I assume that all climate data comes from flux towers (Eddy covariance method?). Conclusions can be extended with recommendations for spatial scale of the application of the proposed approach, e.g. local, regional or global scale and effect of vegetation type since it is mentioned in the manuscript.
Overall impression is good - technical and repeatable study representing straight forward approach for comparison of 2 remote sensing methods using the same set of field measurement data.
Author Response
Response: Thank you very much for your recognition of this paper! With regard to your doubts about this paper, we make the following responses.
(1). Beyond question, the quality of the field measurement data (including the GPP and PAR) are crucial for the model development. FLUXNET 2015 is the widely used data for GPP estimation model design (especially in parameter optimization) and the model accuracy verification. In this paper, sites with high quality data (“NEE_QC” > 0.75) were selected from the FLUXNET 2015 dataset, indicating that at least 75% of high quality daily observations are available during the year.
(2). Management related information in each site is difficult to obtain (FLUXNET 2015 does not provide related information), and such types of variables can cause less effects to this paper’s results. For example, managements (e.g., fertilization and irrigation) in crop are conducive to vegetation growth, which would make it closer to the hypothesis of ‘ideal conditions’ (i.e., unconstrained by nutrients and water). In other words, managements only affect whether the photosynthesis system is under the “ideal condition”. Therefore, such situations would not greatly influence our results, because all sites (no matter under the natural conditions or other artificial conditions) is simply divided into two groups (the ideal condition or not) in our model design.
(3). Given the introduction of such selected sites were not clear enough in the previous version, the vegetation types of selected sites and their corresponding amounts are shown in figure 1 in the revised version. Meanwhile, we also supplemented some information about vegetation types (L94-97).
(4). Shortwave radiation (SR) is one of the basic radiation components that measured as ancillary data for the EC fluxes. Additionally, the daily PAR was calculated using the site observed shortwave radiation according to the empirical formula (i.e., PAR = 0.45 × SR), which is regarded as the benchmark for PAR related inversions (e.g., validation). Therefore, the quality of PAR data is reliable. In the ‘2.1. FLUXNET Data’, we added the corresponding description (L86-90).
(5). According advices, we added the corresponding summary about the application of PAR-LUE in our conclusion (L319-321).
L319-321: “The PAR-LUE was suitable for remote sensing based GPP estimation of most vegetation types at regional and global scales.”
Reviewer 2 Report
This study developed a new method for estimating the LUE used in satellite-based GPP estimation. Considering the spatiotemporal variation of LUE in GPP estimation would be crucial to improve understanding the carbon cycle of the globe, and results of this study look good and are interesting.
However, I have a concern in developing the new model. The problem is that Eq.(8) used to derive the proposed model is physically questionable and Eqs.(8) and (9) are inconsistent from physical point of view. First, FPAR is missing in Eq.(8) while it is required for GPP estimation as shown in Eq.(4). Additionally, in the ideal condition authors stated, GPP_max is represented by a non-linear function of PAR while GPP_i is represented by a linear function of PAR. Thus assumption for Eqs.(8) and (9) is physically inconsistent.
Consideration of the variable LUE would be necessary for GPP estimation, but authors need to preserve physical consistency in developing a new method. I recommend reconsidering the physically valid assumption to develop a method for estimating spatially and temporally dynamic LUE values for more accurate GPP estimation.
For these reasons, current manuscript would not be appropriate for publication.
Author Response
Response:
The common structure of the LUE model can be formulated as GPP = PAR × FPAR × Ɛmax ×f (Ts) × f (Ws). Under the ideal condition that vegetation was unconstrained by other conditions, i.e., FPAR, f (Ts) and f (Ws) are equal to 1. In Eq.(8), the FPAR is not missing, which is assumed to be 1 (L130-132), i.e., GPP = PAR× Ɛmax × FPAR × f (Ts) × f (Ws), where FPAR × f (Ts) × f (Ws) = 1.
On the surface, the Eqs.(8) and (9) seems to be physically inconsistent. However, it is important to note that Ɛmax in Eq (8) is not a fixed value but a PAR related dynamic value. Therefore, GPPi (Eq.(8)) is also represented by a non-linear function of PAR. To further optimize the expression, we revised the Eq.(8) as GPPi = PAR × Ɛmax = PAR × f (PAR), and reorganized the description of PAR-LUE (L128-134, L140).
Round 2
Reviewer 2 Report
Physically inconsistent equations should not be used in developing a method for parameter estimation. Additionally, it is not easy to understand that FPAR is assumed to be 1 in developing method even if it is ideal condition (e.g., LAI is larger than 7).
Thus, several assumptions used in this study is unreasonable.
Author Response
Response to reviewer 2
Reviewer 2
Comments and Suggestions for Authors
Physically inconsistent equations should not be used in developing a method for parameter estimation. Additionally, it is not easy to understand that FPAR is assumed to be 1 in developing method even if it is ideal condition (e.g., LAI is larger than 7).
Thus, several assumptions used in this study is unreasonable.
Response: We appreciate the reviewer’s insightful suggestion and have already amended the descriptions (include equations and words) to clarify the consistency of equations 8 and 9 in the previous round of response.
Here, we will focus on explaining the rationality of assuming FPAR as 1 in the ideal condition. First, the theoretical maximum value of FPAR can reach 1 with respect to the design of remote sensing based estimating formula of FPAR. For example, as for the equation form ref [1]: FPAR= 1 - e-LAI, FPAR = 0.999 ≈ 1 when LAI = 7; and the equation from ref [2]: FPAR = (EVI - 0.1) × 1.25, FPAR = 1 when EVI ≥ 0.9. Additionally, the maximum value of measured FPAR is also close to 1 [3] (Figure 1; cited from [3]). To sum up, the assumption that the FPAR is assumed to be 1 in the ideal condition is reasonable in this paper.
[1] Tao, X., Z. Xiao, and W. Fan, Chapter 11 - Fraction of absorbed photosynthetically active radiation, in Advanced Remote Sensing (Second Edition), S. Liang and J. Wang, Editors. 2020, Academic Press. p. 447-476.
[2] Zhang, Y., et al., A global moderate resolution dataset of gross primary production of vegetation for 2000–2016. Scientific Data, 2017. 4: p. 170165.
[3] Li, W., and Fang, H. (2015), Estimation of direct, diffuse, and total FPARs from Landsat surface reflectance data and ground-based estimates over six FLUXNET sites, J. Geophys. Res. Biogeosci., 120, 96– 112, doi:10.1002/2014JG002754.
